# Optical Nanomotion Detection to Rapidly Discriminate between Fungicidal and Fungistatic Effects of Antifungals on Single-Cell *Candida albicans*

**DOI:** 10.3390/antibiotics13080712

**Published:** 2024-07-29

**Authors:** Vjera Radonicic, Aleksandar Kalauzi, Maria Ines Villalba, Ksenija Radotić, Bart Devreese, Sandor Kasas, Ronnie G. Willaert

**Affiliations:** 1Research Group Structural Biology Brussels, Vrije Universiteit Brussel, 1050 Brussels, Belgium; vjera.radonicic@vub.be (V.R.); ronnie.willaert@vub.be (R.G.W.); 2Alliance Research Group VUB-UGent NanoMicrobiology (NAMI), Vrije Universiteit Brussel, 1050 Brussels, Belgium; Ghent University, 9000 Ghent, Belgium; 3International Joint Research Group VUB-EPFL NanoBiotechnology & NanoMedicine (NANO), Vrije Universiteit Brussel, 1050 Brussels, Belgium; 4Department of Life Sciences, Institute for Multidisciplinary Research, University of Belgrade, 11030 Belgrade, Serbia; 5Laboratory of Biological Electron Microscopy, Ecole Polytechnique Fédérale de Lausanne (EPFL), University of Lausanne, 1015 Lausanne, Switzerland; 6Laboratory of Microbiology, Department of Biochemistry and Microbiology, Ghent University, 9000 Ghent, Belgium; 7Centre Universitaire Romand de Médecine Légale, UFAM, Université de Lausanne, 1015 Lausanne, Switzerland

**Keywords:** cellular nanomotion, single cell, optical nanomotion detection, fungicidal, fungistatic, antifungal, microfluidic chip, microwells, yeast, *Candida albicans*

## Abstract

*Candida albicans* is an emerging pathogen that poses a significant challenge due to its multidrug-resistant nature. There are two types of antifungal agents, fungicidal and fungistatic, with distinct mechanisms of action against fungal pathogens. Fungicidal agents kill fungal pathogens, whereas fungistatic agents inhibit their growth. The growth can be restored once the agent is removed and favorable conditions are established. Recognizing this difference is crucial as it influences treatment selection and infection prognosis. We present a technique based on optical nanomotion detection (ONMD) (i.e., observing the movement of the cells using an optical microscope) to discriminate rapidly between fungicidal (caspofungin) and fungistatic (fluconazole) drugs. The technique is based on the change in a yeast cell’s nanomotion as a function of time during a two-hour treatment with the antifungal of interest followed by a one-hour growth period. The cells are entrapped in microwells in a microfluidic chip, which allows a quick exchange of growth medium and antifungal agent, enabling ONMD measurements on the same individual cells before and after treatment. This procedure permits to discriminate between fungicidal and fungistatic antifungals in less than 3 h, with single-cell resolution by observing if the nanomotion recovers after removing the treatment and reintroducing growth medium (YPD), or continues to drop. The simplicity of the approach holds promise for further development into a user-friendly device for rapid antifungal susceptibility testing (AFST), potentially being implemented in hospitals and medical centers worldwide in developed and developing countries.

## 1. Introduction

*Candida* species cause most human fungal infections, the most frequent ones being *C. albicans*, *C. glabrata*, *C. auris*, *C. tropicalis*, *C. parapsilosis*, and *C. krusei* [1] infections causing 95% of invasive diseases [2]. Candida species are symbiotic in the human gastrointestinal (GI) tract and are common colonizers of the skin, gastrointestinal, and genitourinary tract of 30–70% of healthy individuals. The development of invasive disease generally requires a change in the composition of the competitive commensal bacteria (usually caused using broad-spectrum antibiotics [2]), inability to manage *C. albicans* colonization on mucosal membranes and skin, or the compromising of the patient’s immune system [3]. There are two subtypes of infections caused by *C. albicans*, mucosal and systematic. Mucosal candidiasis primarily affects the vagina, oral cavity, and esophagus. Skin candidiasis is exceedingly uncommon and may rarely occur in a small proportion of patients with certain immunodeficiencies. Systemic candidiasis affects sterile body sites such as the bloodstream, central nervous system (CNS), liver, spleen, heart, and kidneys [4]. The mortality of Candida infections varies across different age groups, but mortality associated with adults is 15% to 20% [5]. Many patients at risk for systemic mycotic infections have chronically weakened immune systems. This includes AIDS patients, who are prone to infections like candidiasis and aspergillosis, requiring ongoing suppressive therapy to prevent relapse; and tissue transplant recipients, who face severe immune suppression post-transplant, leading to potential chronic immune deficiency [6].

Three groups of antifungals are available for treating fungal infections: triazoles (fluconazole, itraconazole, voriconazole, posaconazole, and isavuconazole), polyenes (amphotericin B), and echinocandins (caspofungin, anidulafungin, and micafungin) [5]. Antifungals are classified according to their fungistatic or fungicidal effect. Fungistatic drugs inhibit fungal growth, while fungicidal drugs kill fungal pathogens [6]. Fluconazole, a fungistatic agent, is one of the most prescribed antifungals to treat candidemia. Fluconazole inhibits the growth of the cell by disrupting ergosterol biosynthesis, which leads to the accumulation of methylated sterols in the cellular membrane. This is achieved by inhibition of the lanosterol demethylase (cytochrome P450 enzyme) by binding a free nitrogen atom of the azole ring to the iron atom within the heme group of the enzyme [7]. Fluconazole is widely used to treat candidemia since it has a wide antifungal spectrum. However, its fungistatic characteristics provide an opportunity for the development of fluconazole resistance, especially in immunocompromised patients. There are several mechanisms leading to azole resistance: upregulation of drug transporters, decreased lanosterol 14-*α*-demethylase binding affinity for the drug, increased concentration of lanosterol 14-*α*-demethylase, and inactivation of C5 sterol desaturase leading to alterations in the ergosterol synthetic pathway [8]. Different mechanisms are frequently combined, resulting in a stepwise development of fluconazole resistance over time. Caspofungin is a lipopeptidic antifungal agent that leads to the formation of fungal cell walls without structural integrity, which results in cell vulnerability to osmotic lysis. Caspofungin acts by noncompetitive blockage of the (1,3)-*β*-d-glucan synthase, which leads to the disruption of the cell wall synthesis, therefore, reducing fungal growth or reducing the integrity of the cell wall and reducing its mechanical strength, leading to cell death due to its inability to withstand intracellular osmotic pressure [9].

To determine whether an antifungal agent exhibits fungicidal or fungistatic properties, several common methodologies have been employed. Minimum Inhibitory Concentration (MIC) testing, typically using broth microdilution or agar dilution, is the primary method for identifying the lowest concentration that inhibits fungal growth [10]. However, this is not a good representation of whether the antifungal is truly fungicidal or fungistatic. To determine fungicidal activity, Minimum Fungicidal Concentration (MFC) testing is performed by subculturing samples from MIC assays onto antifungal-free agar plates and identifying the lowest concentration that prevents colony formation [11,12]. Time-kill assays provide insights into the rate and extent of fungal cell death over time by exposing fungal cultures to the antifungal agent and counting colony-forming units (CFUs) at various intervals [13]. Checkerboard assays assess the synergistic, additive, or antagonistic effects of antifungal combinations, potentially indicating fungicidal versus fungistatic interactions [14,15]. Flow cytometry, utilizing viability dyes, differentiates live from dead cells post-treatment. Vital staining and microscopy offer visual confirmation of cell viability [16,17,18]. Lastly, ATP bioluminescence assays measure cellular ATP levels as an indicator of metabolic activity, with decreased ATP levels suggesting reduced viability [19].

Optical nanomotion [20,21,22,23] is a new technique that permits assessing not only live/dead states and transitions of virtually all unicellular organisms on Earth but also monitoring ‘online’ their metabolic state in a label- and attachment-free manner [20]. The technique relies only on a traditional optical microscope, a video camera, and dedicated video processing software. We first applied the ONMD method for antifungal susceptibility testing of pathogenic yeasts such as *C. albicans*, *C. glabrata,* and *C. lusitaniae* [20]. Next, we developed an antifungal susceptibility testing microfluidic chip that contains no-flow yeast imaging chambers in which the growth medium can be replaced by an antifungal solution without disturbing the nanomotion of the cells in the imaging chamber [21]. This allowed recording the cellular nanomotion of the same cells at regular time intervals before and throughout the treatment with an antifungal. Hence, the real-time response of individual cells to a killing compound can be quantified. In this way, this killing rate provides a new measure to rapidly assess the susceptibility of a specific antifungal. It also allowed determining the ratio of antifungal resistant versus sensitive cells in a population. We further improved this technique by designing a microwell-based chip and testing its efficiency [22]. The new device traps cells in microwells and permits the observation of their nanomotion before, during, and after the treatment with a chemical compound. We used ONMD to assess the effect of X-ray radiation on *C. albicans* and its sensitivity to antifungal drugs [24] and demonstrated that exposure to X-ray radiation and radiation in combination with the antifungal fluconazole compromised nanomotion. The nanomotion rate was found to depend on the phase of the cell cycle, the absorbed radiation dose, the fluconazole concentration, and the duration of the post-irradiation period. Furthermore, ONMD has proven to be an efficient and rapid antimicrobial susceptibility testing (AST) method, applicable to motile and non-motile, Gram-positive and Gram-negative, as well as rapid and slow-growing bacteria [23,25].

Here, we developed a new method for fast (less than 3 h) discrimination between fungistatic and fungicidal antifungals using optical nanomotion detection. The new method consists of exposing the yeast cells to the drug of interest for a given period (2 h), replacing it with the culture medium, and observing the nanomotion of the cells for an additional 1 h. Cells exposed to a fungistatic drug recover their nanomotion whereas fungicidal treated cells do not. The effect of caspofungin as a fungicidal and fluconazole as a fungistatic compound was evaluated using the opportunistic yeast *C. albicans* by monitoring their nanomotion pattern as a function of time.

## 2. Results

We observed the effects of a fungicidal and a fungistatic antifungal, as well as ethanol on the nanomotion of *C. albicans* CAF2-1 wild-type strain. For these experiments, we used a microfluidic chip containing microwells [22] in which the cells were trapped (Video S1) and observed. Cells were grown for 60 min and fed into the microfluidic chip to be trapped in the microwells. Next, the growth medium was replaced by the growth medium containing ethanol or the antifungal compound, and the cellular nanomotion was recorded as a function of time. After 120 min exposure to the antifungal, the medium was replaced by the growth medium, and cellular nanomotion was further observed during 60 min. The nanomotion parameter that we obtain is the distance (µm) that the cell “travels” between each frame of the movie called “displacement of the cell”. The length of the movie is 15 s and it is recorded at the frame rate of 10 fps, giving us in total of 150 frames and 149 steps of displacement.

We first evaluated the effect of a rapid killing compound, i.e., 70% (*v*/*v*) ethanol on *C. albicans* CAF2-1 cells over a period of 120 min. After this period, we exchanged the liquid with growth medium YPD and followed the nanomotion for another 60 min. The average displacement (Figure 1a) and displacements per frame (Appendix A) dropped after 60 min of treatment with the killing compound and stayed low for 120 min of treatment. After exchanging the liquid with YPD, the nanomotion did not further increase. To obtain further insight into the effects of fungistatic and fungicidal antifungals, we observed their effects on single cells over time. We followed 20 cells. During the treatment, all cells died after 60 min, and the nanomotion stayed low even after the reintroduction of YPD (Figure 2a). As depicted in Figure 3. the path of cell 1 after removing the treatment did not change (Figure 3a).

In the next experiments, we evaluated the effect of 100 µg/mL of caspofungin on *C. albicans* CAF2-1 cells (i.e., the wild-type strain susceptible to caspofungin). A significant drop in both average (Figure 1b) and displacements per frame (Appendix A) could be observed after 60 min of treatment, and it continued to drop for 120 min. After exchanging the liquid with the growth medium, the nanomotion dropped even further after 60 min of treatment. During the treatment (Figure 2b), we observed that cell 4, cell 11, and cell 20 (Figure 3b) had the largest drop in nanomotion after treating them for 120 min with the antifungal and another 60 min with YPD. The rest of the cells still had a significant and consistent drop in the nanomotion during this period.

Finally, we evaluated the effect of 100 µg/mL of fungistatic fluconazole on *C. albicans* CAF2-1 cells over a period of 120 min, after which we exchanged the liquid with growth medium YPD and followed the nanomotion for another 60 min. A significant drop in both the average (Figure 1c) and displacements per frame (Appendix A) could be observed after 60 min of treatment, and it proceeded to drop for the following 120 min. After exchanging the liquid with the growth medium, the average displacement after 60 min significantly increased. There was no difference between the average displacement 60 min after the treatment, and 60 min after YPD. Following the treatment, we observe a larger drop in all cells even after 60 min of treatment. After exchanging the treatment with YPD, we observe an increase in nanomotion in all cells except cells 8, 11 16, and 17 (Figure 2c). Following the path of the cell across 150 frames for each timepoint, we could observe that the nanomotion dropped during the treatment with fluconazole but increased once we removed the compound and reintroduced the growth medium (Figure 3c).

The slope of the decrease in nanomotion activity, which represents the average displacement per time or the average displacement velocity, provided information on how fast the cell reacted to a specific chemical compound (Figure 1d). With this type of signal analysis, we highlighted the different effects of fungicidal and fungistatic compounds on *C. albicans* CAF2-1 cells. The first data set between the timepoint 0 min and 120 min after treatment represents the average displacement velocity of the cells, which is negative since the nanomotion signal decreased as a function of time. We can observe that the drop in nanomotion happens the fastest during the treatment with ethanol by having the most negative velocity, then fluconazole, and finally with caspofungin. When we reintroduced the growth medium, we followed the trend in the slopes of the decrease in the average displacement of the cells between the 120 min of treatment and 60 min of YPD (Figure 1d). Caspofungin treatment had the most negative velocity, indicating that nanomotion continued to drop, then ethanol, with a slightly negative slope, indicating that the nanomotion dropped by a small amount. Finally, the fluconazole treatment had a positive velocity, since the nanomotion increased.

Fast Fourier Transform (FFT) amplitude spectra analysis of the nanomotion data recorded upon exposure to caspofungin (Appendix A) and fluconazole (Appendix A) treatments, revealed that their levels are correlated with the results obtained using the total displacement analysis (Figure 1). During the caspofungin treatment. The average FFT amplitude dropped as a function of time and continued dropping as we removed the treatment and introduced YPD. Conversely, during the treatment with fluconazole, the average FFT amplitude dropped as a function of time, but after removing the treatment and reintroducing YPD, the average amplitude increased and surpassed the value of the control experiment.

## 3. Discussion

In this study, we evaluated the effects of fungicidal and fungistatic antifungals on *C. albicans* CAF2-1 strain using the optical nanomotion detection technique. This method allowed us to rapidly discriminate between fungistatic and fungicidal actions by monitoring the nanomotion of individual yeast cells within a microfluidic chip. Our results demonstrate that this technique can effectively and rapidly (3 h) distinguish between the effects of caspofungin, fluconazole, and ethanol on *C. albicans*.

The results from the nanomotion analysis indicated that ethanol, known for its strong fungicidal properties, resulted in a rapid and sustained decrease in nanomotion, with no recovery observed after the medium change (Figure 1a), confirming its lethal effect on *C. albicans*. Caspofungin, a fungicidal agent, significantly reduced cellular nanomotion over the 120 min treatment period, and this reduction continued even after the compound was replaced with YPD (Figure 1b). This suggests that caspofungin effectively kills the yeast cells, preventing any recovery of nanomotion activity. In contrast, fluconazole, a fungistatic agent, also caused a significant drop in nanomotion during treatment (Figure 1c). However, by replacing the treatment medium with YPD (Figure 1c), the nanomotion activity of the cells increased, indicating that fluconazole only inhibits fungal growth rather than killing the cells.

The single-cell analysis provided deeper insights into the heterogeneous responses of individual cells to antifungal treatments. Ethanol treatment uniformly decreased nanomotion in all cells (Figure 2a), reinforcing its fungicidal potency. Caspofungin treatment resulted in a consistent decrease in nanomotion across most cells, with a few cells showing a pronounced drop (Figure 2b). Fluconazole treatment, conversely, showed varied responses among individual cells, with some cells recovering nanomotion activity after the medium was replaced (Figure 2c), highlighting the reversible nature of its fungistatic effect.

The averaged displacement velocity (the slope analysis of nanomotion activity over time) further differentiated the effects of the antifungals (Figure 1d). Ethanol exhibited the steepest negative displacement velocity, reflecting its rapid and irreversible killing effect. Fluconazole had a moderate velocity during treatment but a positive velocity after medium replacement, indicating a reversible inhibition of growth. Caspofungin’s velocity remained negative throughout, showing its sustained killing effect even after the antifungal was removed.

Our findings suggest that optical nanomotion detection is a powerful and rapid method for distinguishing between fungicidal and fungistatic antifungal agents. The optical nanomotion detection technique offers several advantages over traditional methods for assessing antifungal effectiveness. It provides rapid results, differentiating between fungicidal and fungistatic effects in less than three hours, compared to the 24 to 48 h required by classical methods such as MFC [11,12] and time-kill assays [13]. This technique also allows for single-cell resolution, giving detailed insights into the heterogeneity of cellular responses, unlike traditional methods that measure population responses, making it easier to detect the potential for developing resistance. The non-destructive nature of optical nanomotion detection allows for the analysis of the same cells, whereas traditional methods often require cell lysis or plating, which prevents further analysis. Additionally, minimal sample preparation is required, and no labeling is needed, reducing potential artifacts and simplifying the experimental workflow compared to techniques such as flow cytometry [16,17,18]. The current setup focuses on single-cell analysis. The advantages of single-cell analysis are that it can detect early signs of drug resistance before they become apparent in the population as a whole. This allows for more timely intervention and adjustment of treatment strategies [26]. Insights from single-cell analysis can lead to more personalized approaches in antifungal therapy, tailoring treatments based on the specific responses of different cells within a patient’s fungal population. Rare but clinically significant phenotypes, such as highly drug-resistant cells, can be identified and studied in detail through single-cell analysis, whereas these might be missed in bulk population studies [27]. Population-level analyses can average out important details and nuances in cellular responses. Single-cell analysis avoids this averaging effect, providing a clearer and more accurate picture of drug efficacy. The disadvantages of single-cell analysis are that it might limit throughput compared to high-throughput screening methods. However, this limitation is addressable, as the average data for the population of cells show promising results. Therefore, it is feasible to scan a larger number of cells to assess whole populations, without relying solely on single-cell assessment.

## 4. Materials and Methods

### 4.1. Yeast Cultivation

For fungicidal and fungistatic susceptibility testing, we used *C. albicans* strain CAF2-1 (genotype: Δura3::imm434/URA3) [28]. Yeast cells were cultured by inoculating a colony from a YPD agar plate (YPD containing 20 g/L agar) with 20 mL of YPD medium (10 g/L yeast extract, 20 g/L peptone, 20 g/L dextrose) overnight in 50 mL Falcon tubes at 30 °C and 180 rpm (Innova 4400, New Brunswick, Edison, NJ, USA). The overnight cultures were then diluted in YPD medium to an optical density at 600 nm (OD_600_) of 0.5 and grown in Falcon tubes for 1 h at 30 °C and 180 rpm. The cultures were further diluted, based on the cell concentration (OD_600_ value), to achieve an optical density of 0.02.

### 4.2. Microfluidic Chip Construction

The microfluidic chip with a microwell layer was fabricated according to the manuscript by Radonicic et al. (2023) [22]. Soft lithography was used to procure the mold for the PDMS membrane layer with microwells of the size 50 × 50 × 50 µm. The microfluidic chip was assembled by attaching six PDMS membranes to a glass coverslip, aligning them with the six channels of a sticky Ibidi µ-Slide VI 0.4 chip, which served as the top plate (Figure 4a,b). Each PDMS membrane contained a 20 × 87 array of microwells (Figure 4c,d).

### 4.3. Microfluidic Chip Setup

A pressure-driven pump (LineUp™ Push-Pull, Fluigent, Le Kremlin-Bicêtre, France) was connected to the inlet and the outlet of the imaging chambers using fluorinated ethylene propylene (FEP) tubing with an internal diameter of 0.51 mm. The microfluidic chip setup was mounted on an inverted Nikon Eclipse Ti2 epifluorescence microscope (Nikon, Tokyo, Japan) to perform ONMD with bright-field microscopy. Initially, the microfluidic chip and tubing were sterilized and air bubbles were removed by flushing with 70% (*v*/*v*) ethanol. Following this, the tubing and chip were flushed with YPD medium to eliminate any residual ethanol. The yeast liquid culture in YPD was then introduced into the channels at a flow rate of 120 μL/min using the pressure-driven pump for 5 min. Following this, the pressure pump was turned off to allow the cells to settle in the wells. The YPD medium was flushed again to remove any excess cells. The first ONMD movie was then recorded and labeled as the “control”. The solution containing the antifungal caspofungin (Sigma-Aldrich, Overijse, Belgium), fluconazole (Sigma-Aldrich, Overijse, Belgium), or 70% (*v*/*v*) ethanol was then filled in from the inlet at a flow rate of 120 μL/min, using the pressure-driven pump for 5 min (until all the liquid in the channel was exchanged with the treatment). This estimate was based on the time required to pump the compound through the tubing with a flow rate of 120 µL/min and the time it took for the compound to reach a maximal concentration at the bottom of the well. The second movie was acquired 60 min after we stopped pumping in the treatment. The next movie was taken 120 min after adding the treatment. Next, the YPD medium was flushed through again at a flow rate of 120 μL/min using the pressure-driven pump for 5 min. The last movie was acquired 60 min after the treatment in the channel was exchanged with YPD.

### 4.4. Nanomotion Measurement and Analysis

Cellular nanomotion was tracked by recording bright-field movies consisting of 150 frames at a frame rate of 15 fps using PCO Edge 4.2 sCMOS camera (Excelitas PCO GmbH, Kelheim, Germany) with a 40× objective, where 1 pixel represented 0.16 µm. The movies were analyzed using the cross-correlation method [29] as previously described [20]. The ONMD algorithm in MATLAB R2022b (MathWorks) was used to compute the X–Y displacement of individual cells in each frame. Displacement distributions per frame were illustrated using violin while the total displacements for all cells were shown with box-and-whisker plots (Prism8, GraphPad).

For each experimental condition and each cell, FFT amplitude spectra were calculated from recorded nanomotions in the X and Y direction, as well as the derived trajectory length time series. The latter was obtained as Euclidean distance between each two elementary cell positions. For each X and Y time series, the total number of frames (points) was 144 (143 for the trajectory length). Since the framing speed was 15 frames/s, all X and Y signal durations amounted to 144/15 = 9.6 s (9.53 s for the trajectory length), resulting in a frequency resolution of 0.104 (0.105) Hz and maximum available frequency of 15/2 = 7.5 Hz. All FFT analyses were performed using our custom MATLAB programs. Amplitude FFT spectra were finally averaged, frequency-by-frequency, for each experimental condition, across the set of individual cells.

## 5. Conclusions

In conclusion, this study highlights the potential of optical nanomotion detection as a novel approach for assessing antifungal efficacy. The ability to rapidly and accurately differentiate between fungicidal and fungistatic agents in three hours could have significant implications for clinical decision-making and the development of new antifungal therapies. Further research is needed to explore the applicability of this technique to a broader range of pathogens and antimicrobial agents.

## Figures and Tables

**Figure 1 antibiotics-13-00712-f001:**
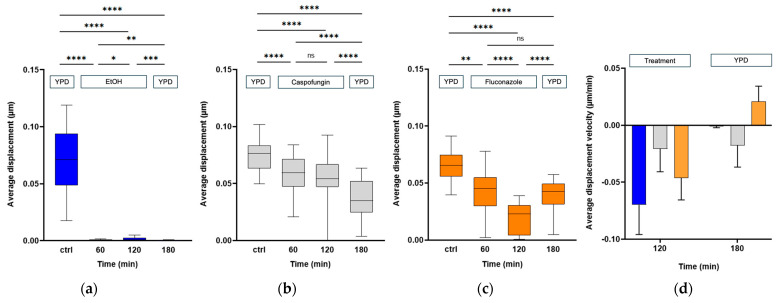
The total displacement of 20 cells was followed during (**a**) 2 h treatment with 70% ethanol (*v*/*v*) followed by 1 h of YPD treatment, (**b**) 2 h treatment with 100 µg/mL caspofungin followed by 1 h of YPD treatment, and (**c**) 2 h treatment with 100 µg/mL fluconazole followed by 1 h of YPD treatment. (**d**) The velocity of the changes in the average displacement of 20 cells as a function of time was calculated based on the time points: 120 min of treatment with ethanol (blue), caspofungin (gray), and fluconazole (orange), and 60 min after replacing the compound with YPD. Wilcoxon test: **** *p* < 0.0001; *** *p* < 0.001; ** *p* < 0.01; * *p* < 0.1; ns: not significant.

**Figure 2 antibiotics-13-00712-f002:**
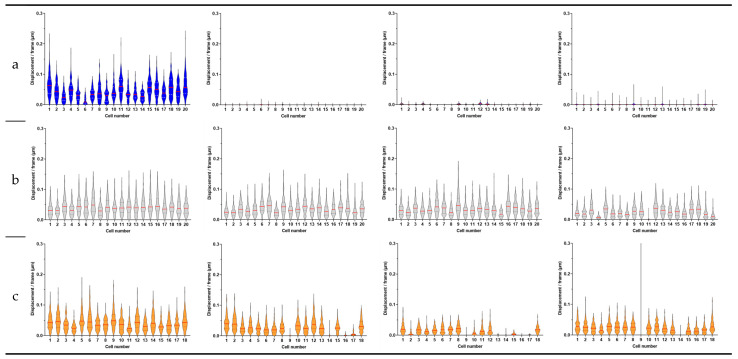
The displacement per frame distribution for 20 single cells of *C. albicans* CAF2-1 wild-type strain displayed as violin plots during (**a**) 2 h treatment with 70% ethanol (*v*/*v*) followed by 1 h of YPD treatment, (**b**) 2 h treatment with 100 µg/mL caspofungin followed by 1 h of YPD treatment, and (**c**) 2 h treatment with 100 µg/mL fluconazole followed by 1 h of YPD treatment. The medians are represented by a red line and the quartiles are represent by black dotted lines.

**Figure 3 antibiotics-13-00712-f003:**
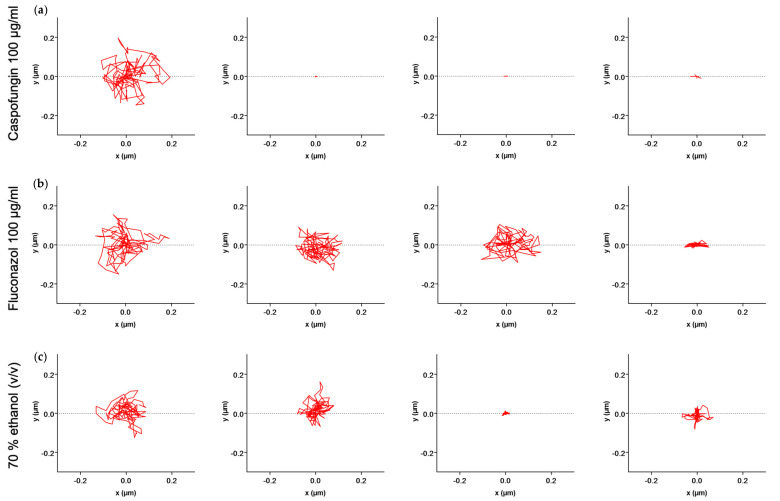
X−Y displacement of (**a**) cell 1 during 2 h treatment with 70% ethanol (*v*/*v*) followed by 1 h of YPD treatment, (**b**) cell 20 during 2 h treatment with 100 µg/mL caspofungin followed by 1 h of YPD treatment, and (**c**) cell 20 during 2 h treatment with 100 µg/mL fluconazole followed by 1 h of YPD treatment.

**Figure 4 antibiotics-13-00712-f004:**
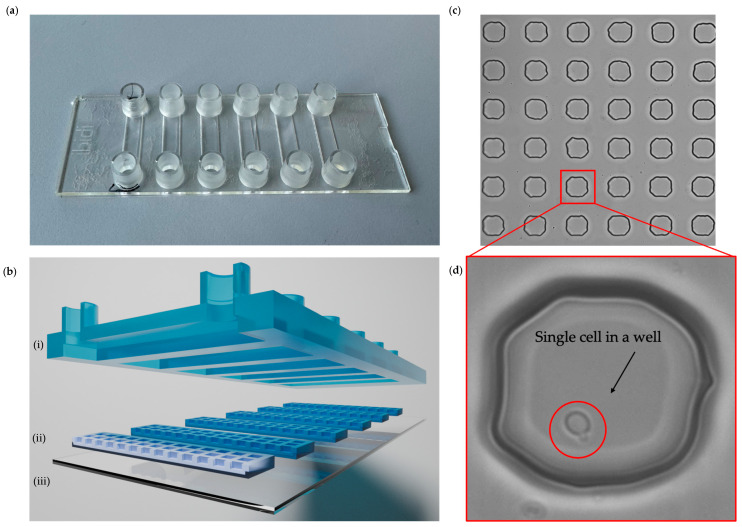
Microfluidic chip with microwells. (**a**) Picture of a microfluidic chip with microwells with six Ibidi channels. (**b**) Schematic cross-section of (**i**) Ibidi chip with channels with inlet and outlet connections, (**ii**) PDMS membranes with microwells size 50 × 50 × 50 µm, and (**iii**) coverslip glass. (**c**) Brightfield image of PDMS membrane in the microfluidic chip. (**d**) Bright-field image of the microwell with a single *C. albicans* cell.

## Data Availability

The original contributions presented in the study are included in the article (and Appendix A), further inquiries can be directed to the corresponding authors.

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
