# Peer review of "Optical Nanomotion Detection to Rapidly Discriminate between Fungicidal and Fungistatic Effects of Antifungals on Single-Cell Candida albicans"

_antibiotics, 2024, doi:10.3390/antibiotics13080712_

Round 1

Reviewer 1 Report

Comments and Suggestions for Authors

In this study, the authors employed a microfluidic chip approach to detect optical nanomotion in single yeast cells and compare the effects of fungicidal and fungistatic substances. However, this technology was introduced and published by the same research team in 2020 (Science Advance), 2022 (Fermentation), and 2023. In their 2023 publication (Fermentation), the applicability of the microfluidic chip was evaluated with fluconazole using the same yeast species; however, in the current study, the authors only extended to caspofungin with a shorter observed duration. It would be helpful if the authors could provide further explanation in the manuscript regarding the originality and logical basis of the current study in relation to their previously published papers.

Overall, the manuscript is well-designed, but the language lacks fluency and ease of comprehension.

Figure 3: legend is not aligned with the figure presented.

Comments on the Quality of English Language

The language necessitated reorganisation. The current version appears to have been translated directly from another language.

Author Response

In this study, the authors employed a microfluidic chip approach to detect optical nanomotion in single yeast cells and compare the effects of fungicidal and fungistatic substances. However, this technology was introduced and published by the same research team in 2020 (Science Advance), 2022 (Fermentation), and 2023. In their 2023 publication (Fermentation), the applicability of the microfluidic chip was evaluated with fluconazole using the same yeast species; however, in the current study, the authors only extended to caspofungin with a shorter observed duration. It would be helpful if the authors could provide further explanation in the manuscript regarding the originality and logical basis of the current study in relation to their previously published papers.

Answer:

To make this clearer the following sentences were added to the Introduction section:

Page 3, line 107: “Optical nanomotion [20], [21], [22], [23] is a new technique that permits assessing not only live/dead states and transitions of virtually all unicellular organisms on Earth but also monitoring ‘online’ their metabolic state in a label- and attachment-free manner [20]. The technique relies only on a traditional optical microscope, a video camera, and dedicated video processing software. We first applied the ONMD method for antifungal susceptibility testing of pathogenic yeasts such as C. albicans, C. glabrata, and C. lusitaniae [20]. Next, we developed an antifungal susceptibility testing microfluidic chip that contains no-flow yeast imaging chambers in which the growth medium can be replaced by an antifungal solution without disturbing the nanomotion of the cells in the imaging chamber [21]. This allowed recording the cellular nanomotion of the same cells at regular time intervals before and throughout the treatment with an antifungal. Hence, the real-time response of individual cells to a killing compound can be quantified. In this way, this killing rate provides a new measure to rapidly assess the susceptibility of a specific antifungal. It also allowed determining the ratio of antifungal resistant versus sensitive cells in a population. We further improved this technique by designing a microwell-based chip and testing its efficiency[22]. The new device traps cells in microwells and permits the observation of their nanomotion before, during, and after the treatment with a chemical compound. We used ONMD to assess the effect of X-ray radiation on C. albicans and its sensitivity to antifungal drugs [24], and demonstrated that exposure to X-ray radiation and radiation in combination with the antifungal fluconazole compromised nanomotion. The nanomotion rate was found to depend on the phase of the cell cycle, the absorbed radiation dose, the fluconazole concentration, and the duration of the post-irradiation period. Furthermore, ONMD has proven to be an efficient and rapid antimicrobial susceptibility testing (AST) method, applicable to motile and non-motile, Gram-positive and Gram-negative, as well as rapid and slow-growing bacteria [23], [25].”

Overall, the manuscript is well-designed, but the language lacks fluency and ease of comprehension.

Figure 3: legend is not aligned with the figure presented.

Answer:

We inverted the row “ethanol 70% (v/v)” and “caspofungin 100 µg/ml” so that the “ethanol” row is on top and the “caspofungin” row is at the bottom and matches the Figure 3. caption.

Comments on the Quality of English Language

The language necessitated reorganisation. The current version appears to have been translated directly from another language.

Answer:

The text was not translated from a different language. The revised manuscript was checked additionally by a native English speaker. The following sentences were adapted:

Page 2, line 47: “Candida species cause most human fungal infections. C. albicans, C. glabrata, C. auris, C. tropicalis, C. parapsilosis, and C. krusei [1] are the most frequent species encountered and they are responsible of 95% of the invasive diseases [2]. Candida species are symbiotic in the human gastrointestinal tract and are common colonizers of the skin and genitourinary tract of 30–70% of healthy individuals.”

Page 2, line 66: “Three groups of antifungals are available for treating fungal infections: triazoles (fluconazole, itraconazole, voriconazole, posaconazole, and isavuconazole), polyenes (amphotericin B) and echinocandins (caspofungin, anidulafungin, and micafungin) [5]. Antifungals are classified according to their fungistatic or fungicidal effect. Fungistatic drugs inhibit fungal growth, while fungicidal drugs kill fungal pathogens [6].”

Page 3, line132: “Here, we developed a new method for fast (less than 3 hours) discrimination between fungistatic and fungicidal antifungals using optical nanomotion detection. The new method consists of exposing the yeast cells to the drug of interest for a given period (2 hours), replacing it with the culture medium, and observing the nanomotion of the cells for an additional 1 hour.”

Page 5, line 227: “After exchanging the treatment with YPD we observe an increase of nanomotion for all cells, except for cells 8, 11 16, and 17 (Figure 2 (c)).”

Page 6, line 248: “Caspofungin treatment resulted in the most negative velocity, indicating that nanomotion continued to drop. Ethanol treatment resulted in a slightly negative slope, indicating that the nanomotion dropped by a small amount.”

Page 6, line 252: “Fast Fourier Transform (FFT) amplitude spectra analysis of the nanomotion data that were recorded upon exposure to caspofungin (Figure S2) and fluconazole (Figure S3), showed that their levels were correlated with the results obtained using the total displacement analysis (Figure 1).”

Reviewer 2 Report

Comments and Suggestions for Authors

The paper entitled:”Optical Nanomotion Detection to Rapidly Discriminate Between Fungicidal and Fungistatic Effects of Antifungals on Single-Cell Candida albicans” by Radonicic et al. deals with a novel optical nanomotion detection (ONMD) system composed by an optical inverted microscope (Nikon Eclipse Ti2 epifluorescence) over a custom microfluidic chip able to entrap individual cells in microwells  and to observe them in motion after exchanging growth medium,  adding biocide agents, etc….

The system is well described and essentially interesting from various points of view. Pictures taken are high quality and even a video is provided as Supplemental material. However, in this work, claims have been made that must be better supported before being released in the scientific literature.

Here is a short list of main concerns:

  1. “rapidly discriminate” .. three hours of observation plus software data analysis is far from being a fast

  2. “discriminate between fungicidal and fungistatic” has been shown only for two molecules

What is the effect of the biocide concentration? The effect of a biocide is a combination of exposure time and dose.. And how can these be studied fast with this method?

  1. How observations in proposed liquid are relevant to real life biocidal activity when a biocide must be used in different matrices and/or environments? In this regard, sampling methods for culture based methods can be used. How sampling can be performed when combined with the proposed technique? Is a culture step necessary, anyway?

  2. diagrams describing cell motions are difficult to understand and essentially of low meaning for eventual users. Can these be implemented with more useful diagrams for microbiologists, lab technicians, and biomedical scientist users?

  3. Cross-validation with other reference techniques could be useful. 

  4. Single cell analysis is invaluable under many aspects but how can single cell analysis be integrated in order to represent the behavior of a cell population, which is relevant for many biological and biomedical consequences linked to the presence of a pathogen cell population more than to that of a single cell?

  5. How could this technique be implemented to support biomedical research?

In general, a Discussion session where limits and advantages of the technique are addressed and compared with the state of the art of the technology and the scientific literature would greatly improve the interest of this paper.

Author Response

The paper entitled: ”Optical Nanomotion Detection to Rapidly Discriminate Between Fungicidal and Fungistatic Effects of Antifungals on Single-Cell Candida albicans” by Radonicic et al. deals with a novel optical nanomotion detection (ONMD) system composed by an optical inverted microscope (Nikon Eclipse Ti2 epifluorescence) over a custom microfluidic chip able to entrap individual cells in microwells  and to observe them in motion after exchanging growth medium,  adding biocide agents, etc….

The system is well described and essentially interesting from various points of view. Pictures taken are high quality and even a video is provided as Supplemental material. However, in this work, claims have been made that must be better supported before being released in the scientific literature.

Here is a short list of main concerns:

“Rapidly discriminate” … three hours of observation plus software data analysis is far from being a fast

Answer:

Standard methods (https://doi.org/10.1128/jcm.39.3.954-958.2001; https://doi.org/10.1016/S0732-8893(02)00525-4; https://doi.org/10.1128/aac.44.7.1917-1920.2000) usually take 24 to 48 hours to obtain data (even more with slow growing cultures) because they rely on the growth of the cells on an agar plate or liquid culture. Taking this into account, our 3-hour method can be considered as a rapid method. The data analysis is semi-automatized, and it’s performed simultaneously during the experiment.

“Discriminate between fungicidal and fungistatic” has been shown only for two molecules. What is the effect of the biocide concentration? The effect of a biocide is a combination of exposure time and dose. And how can these be studied fast with this method?

Answer:

The aim of this article was not to screen among all the fungicidal and fungistatic antifungals, but to demonstrate that ONMD can discriminate the effects of these two types of antifungals. We selected fluconazole and caspofungin since they are amongst the most widely used antifungal drugs. Their fungistatic and fungicidal effects on Candida albicans have been well documented (https://doi.org/10.2147/IDR.S118892; https://doi.org/10.1155/2013/204237; https://doi.org/10.1093/jac/dkg117).

The effect of a biocide can be observed using this method. One microfluidic chip can measure 6 different concentrations simultaneously (it contains 6 separate channels) and it can even be scaled up. Time of exposure can also be shortened because we are not relaying on visible growth to interpret our result but on the decrease in nanomotion signal, which happens soon as several minutes after exposure (https://doi.org/10.3390/fermentation9040365;  https://doi.org/10.3390/fermentation8050195).

How observations in proposed liquid are relevant to real life biocidal activity when a biocide must be used in different matrices and/or environments? In this regard, sampling methods for culture-based methods can be used. How sampling can be performed when combined with the proposed technique? Is a culture step necessary, anyway?

Answer:

Clinical samples from patients, e.g., samples from the mouth, urine, etc., are cultured in liquid growth medium or on an agar plate. To apply our method, cells, e.g., from a swab from the mouth of an infected patient, can be cultured in liquid medium shortly for a 1 to 2 h and then inserted in the microfluidic chip. The cells could also be cultured directly in the chip, right after taking the sample. Dependent on the metabolic state of the cells, a cultivation step during a short period of 1 to 2 h will help in providing an initial maximal nanomotion signal before the antifungal treatment is applied, which could increase the sensitivity of the measurement (difference between living and dead cells).

Diagrams describing cell motions are difficult to understand and essentially of low meaning for eventual users. Can these be implemented with more useful diagrams for microbiologists, lab technicians, and biomedical scientist users?

Answer:

The nanomotion results that we obtain is the distance (µm) that the cell “travels” between each frame of the movie called “displacement of the cell”. The length of the movie is 15 s and it is recorded at the frame rate of 10 fps giving us in total 150 frames and 149 steps of displacement. The larger the displacement length for these steps (= total displacement), the larger the nanomotion signal. In our manuscript we presented the displacement of yeast cells in different ways to highlight specific characteristics of the nanomotion.

Box plots clearly show the difference between the fungistatic and fungicidal effects (Figure b and c, the last columns clearly display different behavior of yeast following the exposure to fungicidal or fungistatic drugs). We represented the nanomotion displacements also with violin plots since previous studies demonstrated peculiar shapes that depend on the virulence of the microorganism (https://doi.org/10.3390/microorganisms9081545). Finally, nanomotion trajectories were also presented to show the two-dimensional paths of the cells upon exposure to the drugs.

We also included graphs showing the change of the average velocity (displacement/time) of the cells. We believe that these graphs complement each other, and should be useful for scientists as well for clinicians.

To make this clearer the following sentences were added to the Results section:

Page 3, line 149: “The nanomotion results that we obtain is the distance (µm) that the cell “travels” between each frame of the movie called “displacement of the cell”. The length of the movie is 15 s and it is recorded at the frame rate of 10 fps giving us in total 150 frames and 149 steps of displacement.” (135-139)

Cross-validation with other reference techniques could be useful. 

Answer:

The fungistatic behavior of fluconazole and fungicidal of caspofungin is well described in the literature and established using different techniques. By evaluating our method, using the same compounds we could easily confirm that fluconazole is fungistatic and caspofungin is fungicidal. In addition, we included in our study 70% ethanol as a fungicidal compound as a control. Additionally, we included in the study a control experiment before each treatment, observing nanomotion of the cells in YPD. Previously (https://doi.org/10.3390/fermentation9040365), we also observed the nanomotion of the cells in YPD during a 3-hour period.

Single cell analysis is invaluable under many aspects but how can single cell analysis be integrated in order to represent the behavior of a cell population, which is relevant for many biological and biomedical consequences linked to the presence of a pathogen cell population more than to that of a single cell?

Answer:

Single-cell analysis for our method is advantageous since we only need a limited number (i.e., 20) of cells to evaluate if a compound is fungicidal or fungistatic. Additionally, clinical sample size is sometimes restricted. Our method is not restricted to only single-cell analysis but can also be extended to population analysis. In this case, a much higher number (1000-5000 cells) of cells could be analyzed by automating the image acquisition and data analysis (https://doi.org/10.3389/fmicb.2024.1328923; https://doi.org/10.1073/pnas.2221284120).

Both can be done as was addressed in the Discussion: “However, this limitation is addressable, as the average data for the population of cells shows promising results. Therefore, it is feasible to scan a larger number of cells to assess whole populations, without relying solely on single-cell assessment.”

To make this clearer the following sentences were added to the Discussion section:

Page 7, line 309: “Advantages of single-cell analysis are that early signs of drug resistance can be detected before they become apparent in the population as a whole. This allows for more timely intervention and adjustment of treatment strategies. Insights from single-cell analysis can lead to more personalized approaches in antifungal therapy, tailoring treatments based on the specific responses of different cells within a patient’s fungal population. Rare but clinically significant phenotypes, such as highly drug-resistant cells, can be identified and studied in detail through single-cell analysis, whereas these might be missed in bulk population studies. Population-level analyses can average out important details and nuances in cellular responses. Single-cell analysis avoids this averaging effect, providing a clearer and more accurate picture of drug efficacy. The disadvantage of single-cell analysis is that it might limit throughput compared to high-throughput screening methods.”

How could this technique be implemented to support biomedical research?

Answer:

The technique can be implemented for antimicrobial drug discovery and antimicrobial susceptibility testing. One of the main advantages of the technique relies in its simplicity. Traditional optical microscope equipped with a camera and dedicated microfluidic chip are enough to conduct the described experiments. Therefore, it could be easily and widely implemented.

In general, a Discussion session where limits and advantages of the technique are addressed and compared with the state of the art of the technology and the scientific literature would greatly improve the interest of this paper.

Answer:

Thank you for the highly constructive comments. The advantages and limits of the new technique are already included in the Discussion section. The advantages of single-cell analysis were additionally added in this section (see comment before, Page 7, line 312).

Round 2

Reviewer 2 Report

Comments and Suggestions for Authors

Authors have provided answers to our previous criticisms. By adding just a few sentences they have better explained the experimental settings which are now more understandable at a glance by readers. Although validation of the method is still far from being comprehensive, the paper provides a sound initial experimental demonstration of their technique.